# Ultrasound Elastography as a Diagnostic Tool for Peyronie’s Disease: A State-of-the-Art Review

**DOI:** 10.3390/diagnostics14060665

**Published:** 2024-03-21

**Authors:** Gianni Paulis, Giovanni De Giorgio, Andrea Paulis

**Affiliations:** 1Department of Urology and Andrology, Peyronie’s Care Center, Castelfidardo Clinical Analysis Center, 00185 Rome, Italy; 2Section of Ultrasound Diagnostics, Department of Urology and Andrology, Castelfidardo Clinical Analysis Center, 00185 Rome, Italy; g.degiorgio@analisiclinichecastelfidardo.it; 3Neurosystem for Applied Psychology and Neuroscience, Janet Clinical Centre, 00195 Rome, Italy; andrea.fx.94@gmail.com

**Keywords:** elastography, ultrasound, shear-wave imaging, strain imaging, Peyronie’s disease, state-of-the-art review

## Abstract

Elastography is a noninvasive method that utilizes ultrasound imaging to assess the elasticity and stiffness of soft tissue. Peyronie’s disease (PD) is a chronic inflammatory condition that affects the male penis, causing the formation of fibrous plaques. This alters the penis’s elasticity and can lead to changes in its shape. Ultrasound elastography (UE) is an important advancement in the diagnosis of PD. It not only identifies plaques, but it also measures their rigidity, providing crucial information to monitor changes during and after treatment. We conducted a narrative review of the scientific literature to identify articles that discuss the use of elastography in the diagnostic study of PD. The purpose of this study was to describe the “state of the art” in the diagnostic use of ultrasound in combination with elastography to highlight any benefits in the diagnosis of PD. We found 12 relevant articles after searching PubMed, Embase, and Google Scholar using the keywords “ultrasound elastography” and “Peyronie’s disease”, including eight clinical studies, two case reports, and two review articles. The results of our review indicate that UE is a useful technique for identifying Peyronie-related plaques, particularly when they are not detectable using a standard ultrasound or physical examination. It is also helpful in monitoring improvements during and after conservative treatments. More research is required to confirm the effectiveness of ultrasound elastography in diagnosing Peyronie’s disease and to determine whether it is better than traditional ultrasound.

## 1. Introduction

### 1.1. Peyronie’s Disease (PD)

PD is a chronic inflammatory disease that affects the tunica albuginea of the penis in male individuals with a genetic predisposition, leading to the formation of fibrous plaques. This alters a penis’s elasticity and can also lead to changes in its shape [1,2].

The prevalence of PD is higher in the Western world (3.2–13.1%) and lower in Asian countries (0.6–5.0%) and among Black African origin populations (0.1–3.5%). The causes of these differences are not entirely clear, but they may be related to genetic, environmental, and lifestyle factors [3,4,5,6,7,8,9,10,11,12]. PD mostly affects patients in the fifth decade of life, with an average patient age of 55 years. Multiple studies have examined the occurrence of PD in young men, with reported prevalence rates ranging from 1.5% to 10.8% [13,14,15,16]. Some authors report an increase in the prevalence of PD in young people, noting that 18.6% of PD patients were ≤40 years old and 81.4% were >40 years old [17]. In our own study, we found similar results, with 16.9% of PD patients being ≤40 years old [18].

PD can lead to penile deformation, penile pain, erectile dysfunction, and psychological distress; penile deformations may include curvature, shortening, torsion, divots, and hourglass deformity [19,20,21,22,23]. The exact cause of PD is not fully understood, but it is generally believed to be triggered by local injury [24,25,26]. This initial injury, whether minor or major, leads to the deposition of fibrin and the formation of a small hematoma. In individuals not genetically susceptible to the disease, the hematoma is typically reabsorbed. However, in individuals with a genetic predisposition, the hematoma causes local recruitment of inflammatory cells and pro-inflammatory cytokines, resulting in the formation of chronic inflammatory tissue that subsequently progresses to fibrosis [27,28,29]. Over the past 20 years, research has shown that oxidative stress (OS) plays a critical role in plaque formation and in the progression of PD [28,29,30,31,32,33,34].

A diagnosis of PD involves a physical examination, penile palpation, photographic documentation of the curvature, and dynamic color Doppler ultrasound imaging of the penis, as well as computed tomography (CT) and magnetic resonance imaging (MRI) [35,36,37]. Furthermore, elastography has recently been proposed as a diagnostic tool for this disease.

### 1.2. Elastography

Elastography is a noninvasive method used in tandem with an ultrasound probe to measure the mechanical properties of tissue that may be more rigid due to alterations caused by a pathological process. The first diagnostic approaches with elastography began at the end of the last century and were followed by numerous scientific studies [38]. Elastography is mainly used for soft tissue pathologies, such as tissues of the liver, breast, prostate, thyroid, pancreas, nerves, tendons, and muscles. More recently, elastography has also been proposed for the study of PD [37,38,39,40,41,42,43,44,45,46,47,48]. Commercial elastography equipment was initially introduced in 2003 and has seen significant advancements over the past two decades. Currently, many ultrasound manufacturers provide elastographic features, such as built-in capabilities for strain- or shear-wave-based methods. The World Federation of Ultrasound Medicine and Biology (WFUMB) categorizes elastography methods into four groups: strain elastography (SE), transient elastography (TE), acoustic radiation force impulse (ARFI), and shear-wave speed measurement and imaging [49,50,51,52,53,54,55]. There are two main studied techniques for measuring the mechanical properties of tissues: static (or quasi-static) methods and dynamic methods. These techniques use different excitation modes (external forces). A thorough introduction to the principles and techniques of ultrasound elastography is essential for understanding this diagnostic method, which this review aims to provide. Subsequently, this technique’s clinical applications for PD are discussed.

### 1.3. The Aim of this Study

The objective of our study was to perform a narrative review of the scientific literature to search for articles that concern the use of elastography in the diagnostic study of Peyronie’s disease. We describe the “state of the art” of ultrasound and elastography methods for diagnosing PD and highlight any diagnostic benefits. Furthermore, the main objective of this study, conducted through a literature review, was to evaluate the diagnostic efficacy of elastography in patients with Peyronie’s disease and to ascertain whether its integration with conventional ultrasound examination improves diagnostic accuracy, making it a valuable tool for monitoring PD patients undergoing conservative therapy.

## 2. Relevant Sections

### 2.1. Strain Elastography

SE assesses tissue deformation by applying manual compression or exploiting physiological motion. SE is sometimes referred to as real-time elastography (RTE). Initially, several experiments were performed to confirm the validity of this basic method [50,56]. The strain induced using “quasi-static” methods, such as manual compression or cardiovascular/respiratory pulsation, is calculated, and the distributions of the strain or normalized strain values within the region of interest (ROI) are shown. SE is a qualitative technique that measures the relative stiffness of different tissues and analyzes their ability to deform under the action of external forces and, subsequently, return to their original shape. SE shows the deformation along the longitudinal axis. The absolute value of the strain along the compression axis is proportional to the magnitude of the initial compression. Tissue stiffness is quantified by Young’s modulus (E), which is determined by the formula E = s/ε, where s represents stress and ε represents strain.

Deformation is induced by manually applying micropressure with the ultrasound probe or by means of ultrasound pulses of appropriate intensity [49,51,52,53,54,55]. In the second excitation method, the ultrasound transducer remains stationary, while tissue displacement is caused by internal physiological movements (e.g., cardiac and arterial pulsations and respiratory motion). Because superficial compression cannot be easily transmitted to deep organs, the second method is preferred. In both cases, the ultrasound images before and after the compression are then compared, and the equipment generates a color map that expresses the relative elasticity of the different tissue components, finally showing a specific “deformation index”. Usually, blue is used to represent low strain (i.e., stiff tissue), and red is used to represent high strain (i.e., soft tissue), although the specific color may differ based on the manufacturer of the ultrasound device [40,52,54]. SE assesses the relative stiffness of tissues within the elastographic ROI superimposed on a B-mode ultrasound image, requiring precise alignment of the ROI with the Peyronie’s plaque.

Figure 1 displays images from a strain elastography (SE) study performed on two of our patients with Peyronie’s disease. The two cases were examined using Philips Affinity 70G ultrasound equipment and a high-frequency linear transducer, namely the Philips Linear Probe L12-5 (Philips, Washington, DC, USA).

The color map is represented in real time, superimposed on an ultrasound image of the tissues under examination. The presence of a strain “defect” along the compression axis affects all other strains along that axis, so an increase or decrease in the strain of a defect is “distributed” along the axis. Since elasticity is a fundamental property of tissue, this parameter is, ultimately, considered very reliable.

### 2.2. Transient Elastography

Transient elastography (TE) is a dynamic, noninvasive, and nonimaging method for evaluating liver fibrosis by measuring liver stiffness (LS). TE generates a shear wave via external vibration. An external punch with controlled vibration is used to generate shear waves, the average speed of which is measured within a specific ROI and then converted to a Young’s modulus value.

The first shear-wave technique used in clinical practice was transient elastography (TE) with the introduction of the FibroScan^®^ system (Echosens, Paris, France). However, the available equipment for performing TE specializes in measuring tissue stiffness but not imaging [49]. FibroScan was first used in 2003 as a new method to evaluate liver fibrosis noninvasively and as an alternative to liver biopsy [57]. FibroScan systems can be used to evaluate liver fibrosis in various diseases, such as nonalcoholic fatty liver disease (NAFLD), liver cirrhosis, and liver failure. Pulse–echo ultrasound is used to track the shear wave’s propagation and measure its velocity, which is directly related to tissue stiffness. The elastic modulus (E) is expressed as E = 3ρV^2^, where V is the shear velocity, and ρ is the mass density (constant for tissues). This method is very useful for the management and treatment of liver diseases, including antiviral therapy for hepatitis C or for the management of complications related to cirrhosis. In some cases, FibroScan can also be used for nonhepatic fibrous diseases, such as chronic kidney disease. However, for fibrotic kidney disease, the lack of an image overlay on an ultrasound image makes it difficult to use for renal assessments. Furthermore, the few published studies using TE on the kidneys report that the results may be influenced by the heterogeneous morphology of the kidneys [58,59,60].

### 2.3. Acoustic Radiation Force Impulse

Acoustic radiation force impulse (ARFI) is a method of dynamic elastography. In the technical literature, the term “ARFI imaging” is often used interchangeably with ARFI, referring to the same method. However, in clinical literature and commercial products, ARFI is used to describe both ARFI imaging and quantitative ARFI.

ARFI utilizes acoustic radiation force to generate shear waves [49]. In ARFI imaging, focused pulses of acoustic radiation deform the tissue, and the resulting tissue displacement is measured within the focal region of each push in a specified region of interest (ROI). The distribution of the displacement or its normalized values within the ROI is then displayed.

ARFI imaging, on the other hand, is a type of elasticity imaging that utilizes acoustic radiation force impulse (ARFI) excitation to generate images displaying tissue displacement within the ARFI excitation beam. By sequentially examining neighboring lateral positions within a designated field of view using ARFI excitations, the resulting images display relative tissue displacement. These images provide data comparable to strain images produced with external pressure, allowing for ultrasound imaging without blind measurements, unlike transient elastography.

ARFI techniques can be classified into point shear-wave elastography (SWE), 2D SWE, and 3D SWE techniques [54].

#### 2.3.1. Point Shear-Wave Elastography (p-SWE)

P-SWE (or ARFI quantification) measures the speed of shear waves [50,55]. It is a dynamic ultrasound elastography method that measures tissue stiffness. It generates quantitative data, expressed in m/s, related to the tissue portion included in a region of interest of approximately 1 cm^3^ in volume, selected from the B-mode ultrasound image. Point shear-wave elastography (p-SWE) technology was developed and introduced by Siemens AG industry in its Acuson S2000 (ARFI Virtual Touch™, Siemens Healthineers, Erlangen, Germany) ultrasound machine. The Acuson S3000 ultrasound machine is also available as a further evolution of the machine. Another machine that uses the pSWE technique is currently available on the medical market: ElastPQ (Philips Healthcare, Andover, MA, USA). With p-SWE, an acoustic pulse is used to generate transverse shear waves at a point within the liver tissue. An advantage of p-SWE is that it displays an anatomical ultrasound image at the same time, allowing for the correct area to be evaluated. The stiffness of the underlying tissue is correlated with shear-wave velocity. Tissue elasticity is measured starting from a B-mode image. After setting the device to the ARFI mode, the region of interest (ROI), which measures 6 mm long and 10 mm wide, is positioned in the area in which the measurement will be performed. An estimate is then performed by pressing a dedicated button. This operation determines the excitation of the ROI with an impulsive force that generates shear waves within the tissue. The velocity of the shear waves is related to the square root of the tissue’s elasticity. The result is the identification of the stiffness of the area under examination, reported as the speed in meters per second. Limitations of the ARFI technique include its lack of elastic tissue mapping, its inability to perform multiple acquisitions in different regions simultaneously, and the limited depth positioning of the ROI, which are all due to the probe’s limitations [61].

#### 2.3.2. Shear-Wave Elastography

Shear-wave elastography (SWE) is a technique used to excite tissue to generate shear waves and measure shear wave velocity. It provides an estimated value in either shear-wave speed (m/s) or Young’s modulus (kPa) [49,51,52,53,54,55]. SWE enables the real-time, two-dimensional, quantitative imaging of tissue elasticity in combination with traditional ultrasound grayscale imaging. It displays the velocity of a perpendicular wave along the transverse axis. SWE is a dynamic, quantitative method that measures tissue stiffness. In 2D SWE, the intensity of the acoustic radiation is generated from various points, and the time of arrival of the shear wave is then measured sequentially using a flight time estimation method. This allows for the generation of a shear-wave image of a larger region of interest, which can be encoded in either color or grayscale. This type of image can be shown either independently or superimposed on the conventional ultrasound image [54].

The medical companies that produce this 2D SWE equipment are Siemens (Erlangen, Germany) and SuperSonic Imagine S.A. (Aix-en-Provence, France) [54]. Currently, SuperSonic Imagine S.A. is the only medical company developing 3D SWE equipment [54]. The 3D probe developed by this manufacturer includes a series of mechanical 2D scanners and has advanced computing power, enabling it to accurately define tissue stiffness in three dimensions. Three-dimensional SWE technology is currently undergoing further refinement, so recommendations for its use cannot be given at this time. However, a recent article was published in which a 2D linear transducer (L7-4, Philips Healthcare, Andover, MA, USA) was used to generate shear waves through the thrust beam to then be detected by the row–column-addressing (RCA) array and thus produce a 3D shear-wave elastography (SWE) image [62].

The procedures for ARFI-based techniques are the same as those for TE, using the same transducer as for conventional ultrasound. Liver examinations are more commonly performed using a convex probe rather than a linear probe.

Two-dimensional SWE has higher accuracy in diagnosing severe fibrosis than TE (F ≥ 3), as well as a higher accuracy than p-SWE in diagnosing significant fibrosis (F ≥ 2) [54]. There were no notable variances among the three techniques (TE, p-SWE, and 2D SWE) in diagnosing mild fibrosis or cirrhosis [54]. Regarding the study of liver disease, limited data are available on the usefulness of ARFI techniques in predicting prognosis [54]. Liver stiffness (LS) measurements using p-SWE and 2D SWE in this field have shown applicability and diagnostic accuracy similar to those of TE in the available studies [54,61]. In liver studies, ARFI technology has a diagnostic accuracy comparable to transient elastography but a lower failure rate, especially in obese patients. Studies have explored ARFI-based elastography for detecting and characterizing focal liver lesions but have found that it struggles to distinguish between benign and malignant lesions due to the presence of overlapping values [54]. The limited depth of penetration poses challenges to identifying lesions using this technique. ARFI techniques show superior diagnostic performance for thyroid nodules compared to SE, with a higher optimal cut-off value [54,63,64].

#### 2.3.3. Supersonic Shear Imaging

Supersonic shear imaging (SSI) is an elasticity imaging method that uses 2D transient elastography technology but replaces the vibrator with acoustic radiation pressure. The excitation imaging technique is seamlessly integrated into just one component: the transducer array for ultrasound imaging.

The shear wave generated in this technique, amplified by the Mach cone (this term is used because the wave front has a cone-shaped profile), has an amplitude equal to tens of microns. This can be identified using an ultrasound speckle-tracking algorithm and ultrafast imaging with a good signal-to-noise ratio. Using ultrafast imaging, shear-wave propagation can be fully captured in less than 30 ms. The technique is somewhat sensitive to patient movements, such as breathing, and can be displayed in real time, just like a traditional ultrasound image.

Young’s modulus maps are reconstructed in SSI by estimating the velocity of the shear wave between two points in the image using a time-of-flight algorithm. This technique has been employed using the Aixplorer^®^ ultrasound imaging device (Supersonic Imagine, Aix-en-Provence, France), mainly in the study of breast cancer [65]. 

#### 2.3.4. Vibro-Elastography and Vibro-Acoustography

Vibro-elastography and vibro-acoustography are both medical imaging techniques that utilize vibrational energy to assess tissue stiffness. However, there are some notable distinctions between the two, as outlined below.

Vibro-elastography (VE) is an innovative imaging technique that measures the mechanical properties and stiffness of tissues by applying a mechanical force or vibration to the body’s surface using a probe or device. VE utilizes a dynamic excitation source to induce tissue motion across various frequencies. This method analyzes tissue movement in response to a multifrequency external vibration source (mechanical force or vibration). The resulting load response is then analyzed to determine tissue stiffness. Ultrasound images are used to record tissue movement at different locations and time intervals. In a VE version, the tissue is depicted as a network of interconnected mass–spring–damper elements. The density, stiffness, and viscosity parameters are determined by solving a series of equations of motion. In a different iteration of VE, the tissue characteristics are directly derived from transfer functions between specific tissue areas. In both scenarios, the VE images exhibit superior quality (lower standard deviation compared to the mean values) in comparison to static elastography, mainly due to the inherent averaging across different levels and rates of tissue strain.

Vibro-elastography is primarily used to assess fibrosis or the presence of tumor masses in soft tissues, such as in the liver, prostate, or lymph nodes. Additionally, its dynamic approach also allows for the measurement of the viscosity and density [66].

Vibro-acoustography is an imaging technique that uses low-frequency ultrasound to generate acoustic waves in body tissue. These acoustic waves cause mechanical deformation of the tissue, which can be detected and measured. Vibro-acoustography is an elastography method that uses ultrasound radiation pressure and was created by an American research team led by James Greenleaf [67]. This method uses the acoustic radiation force, but, unlike vibro-acoustography, ARFI uses a single beam of focused ultrasound. Radiation pressure is the force per unit volume generated by the transfer of momentum within the medium. This momentum is associated with the absorption of the ultrasound wave. Vibro-acoustography involves the use of two confocal ultrasound beams with slightly different frequencies: ω0 and ω0 + ω. 

This leads to beats occurring at the frequency ω, generating a modulated force at this frequency specifically at the focal point.

Consequently, it appears as though the target is vibrating at the frequency ω. The sound created by this stimulation indicates the mechanical characteristics of the object, specifically its stiffness. To achieve this, a hydrophone can be positioned to capture the target’s response at the frequency ω. The process involves sweeping the whole zone by adjusting the focal point and recording the response at each point to generate an image [53]. The use of vibro-acoustography has been researched in various organs, with a focus on arteries, breast tissue, and the prostate [68,69,70]. 

### 2.4. Literature Review

The databases used in the literature review were PubMed, Embase, and Google Scholar, with “ultrasound elastography” and “Peyronie’s disease” as the keywords. We excluded experimental studies that did not involve male humans. 

For completeness, as well as to satisfy our curiosity, we also carried out a more general search in Medline using only the keyword “ultrasound elastography”.

We found twelve articles, including eight clinical studies, two case reports, and two review articles, after conducting a search in Medline using the keywords “ultrasound elastography” and “Peyronie’s disease” (Figure 2).

Overviews of the 12 published articles involving only male subjects that we found during our research are provided below. They are sorted by ascending year of publication and type, in the following order: studies on cohorts of patients with PD, case reports, and reviews. We also report that our more general and less specific search using Medline for the term “ultrasound elastography” produced 16,032 articles.

#### 2.4.1. Lahme et al. (2009) [38]

In this pioneering study, which was the first of its kind, real-time elastography (RTE) was used in patients with PD [38]. The published article refers to a presentation given at the American Urological Association’s 2009 Annual Meeting (Chicago, IL, USA). The authors conducted a study in which 37 patients with PD were examined and underwent conventional ultrasound (CU) and real-time elastography (RTE). The presentation of the study did not indicate the machinery that was used. The results of the study were as follows: All patients had palpable plaques. In 32.4% of the cases, no plaques were detected using CU, but all palpable plaques were visible using RTE. While CU only showed partially or extensively calcified plaques, RTE allowed for the measurement of the size and thickness of the plaques in each case, both in transversal and longitudinal examinations. The authors concluded that RTE is a promising new imaging technique for patients with PD, which, at the time, had not been previously reported. The authors reported that RTE is reliable for the detection of both palpable and non-palpable plaques, as well as for determining the plaques’ size and thickness. Because RTE is performed simultaneously with CU, no additional time or costs are required to examine the plaques.

#### 2.4.2. Morana et al. (2010) [39]

In this study, 45 patients with PD were examined using real-time sonoelastography to assess penile conditions during flaccidity and drug-induced erections (10 mg PGE1) [39]. The Hitachi Logos Vision (Hitachi Medical System, Tokyo, Japan) ultrasound machine, with second-generation elastosonography equipment, was used in the study. The researchers evaluated the position, length, thickness, and involvement of the surrounding tissue, as well as the septum intercavernosum and rigidity of the plaques.

The results of the study were as follows: Elastosonography detected all 59 plaques present, whereas traditional ultrasound only identified 26 plaques. Additionally, elastosonography found greater lengths, widths, and involvement of surrounding tissues compared to conventional ultrasound. The authors also observed no significant differences in the plaques’ characteristics between erect and flaccid states using sonoelastography. In five cases, traditional ultrasound revealed the involvement of the septum intercavernosum, whereas real-time elastosonography identified eight such cases. Fibrosis of the corpora cavernosa’s underlying plaque was detected in 3 cases using conventional sonography and in 13 cases using real-time elastosonography. The authors concluded that real-time elastography is a highly reliable imaging technique that can detect palpable plaques and accurately assess the thickness, size, and involvement of surrounding tissue and the intercavernous septum.

#### 2.4.3. Riversi et al. (2012) [40]

The authors conducted a study involving 74 patients with PD and a control group consisting of 15 healthy male subjects [40]. The study compared conventional ultrasound images of the penis with the elastosonographic images for each patient to assess the diagnostic performance of this new imaging modality. 

The authors captured conventional ultrasound and real-time elastosonography (RTE) images of the penis in a single session. Two experienced radiologists evaluated the images using a Hitachi EUB 8500 Logos digital US scanning system (Hitachi Medical System, Tokyo, Japan) and a high-frequency linear array transducer (14-6 MHz EUP-L53, Hitachi Medical System, Tokyo, Japan). Real-time freehand US elastosonographic measurements were taken with the same instrument and probe, ensuring optimal adherence to the penis and uniform compression. A standardized freehand compression technique was used, graded on a numeric scale (1–5) to maintain an intermediate level for optimal evaluation. CAM software (first version, patent No. US 8041415 B2, Hitachi Medical Corporation, Tokyo, Japan) was used to provide fine-grained estimation of tissue displacement and generate elastosonographic images, with elasticity values color-coded from red (elastic tissue) to blue (anelastic tissue) and green for average levels of strain. These images were overlaid on corresponding B-mode scans to correlate strain distribution with the ultrasound image.

In healthy men, the elastosonogram of the penis showed a consistently green area of interest, similar to the surrounding tissue. Among the 74 patients with PD, penile plaques were detected in 64 patients (86.49%) using B-Mode US. In the elastosonographic assessment, these plaques appeared as blue areas (anelastic tissue) surrounded by softer green and red tissue. Furthermore, in 7 out of these 64 patients (10.94%), elastosonography revealed a larger, well-defined blue area, indicating a dorsal plaque that was not fully captured by conventional US due to the presence of isoechogenic areas in the tunica albuginea. In the remaining 10 patients with PD, B-Mode US did not show any signs of penile plaque or thickening of the tunica albuginea.

However, in five cases, elastosonography showed a prominent and well-defined blue area, suggesting an area of reduced tissue elasticity, as seen in the presence of fibrosis. The authors concluded that RTE is a novel, noninvasive method that can enhance B-Mode ultrasound in detecting and distinguishing penile plaques in PD patients. They also suggest that it can be helpful in the identification of areas of reduced tissue softness even when penile plaque is not present.

#### 2.4.4. Zhang et al. (2018) [41]

The authors conducted a study involving ten patients: three with PD, five with both PD and erectile dysfunction (ED), and two with isolated ED [41]. Ultrasound vibro-elastography was performed. The main objectives of this study were to evaluate the elasticity and viscosity of corporal tissue before and after the use of erectogenic medication at frequencies of 100, 150, and 200 Hz.

The secondary objectives involved assessing the relationships between erectile function measures and viscoelasticity at baseline and after injection, as well as the change pre- and postinjection.

The penis was subjected to a 0.1 s vibration using a handheld shaker indenter. The Verasonics V1 ultrasound device (Verasonics, Inc., Kirkland, Washington, USA) was utilized with an L11-4 ultrasound probe with a central frequency of 6.4 MHz. The tissue motion in response to the vibration excitation at 100, 150, and 200 Hz was detected using a high pulse repetition rate of 2000 frame/s. After the initial assessments, the patients received intracavernosal injections of a vasoactive trimix (10 μg/mL of alprostadil, 30 mg/mL of papaverine, and 1 mg/mL of phentolamine). The erection was reassessed 15 min later, and additional medication was given until a complete and rigid erection was achieved.

Afterwards, repeat UVE assessments were carried out similar to the initial tests, followed by a traditional penile ultrasound. Although patients with PD were included in the study, the main focus was on tissue variations (i.e., in viscoelasticity) before and after the injection of vasoactive erectogenic drugs to provide a more comprehensive diagnosis of ED. The researchers assessed penile elasticity in patients with ED and PD before and after inducing an erection with vasoactive trimix. They discovered a significant rise in viscoelasticity in patients with induced erections, suggesting that SWE could serve as a valuable tool for tracking changes in erection dynamics.

However, this study did not address the examination of penile plaque rigidity and its specific diagnostic implications as a result of the use of vibro-elastography.

#### 2.4.5. Trama et al. (2018) [42]

The authors carried out a study on 40 patients with Peyronie’s disease using elastosonographic imaging and employing the shear-wave elastography (SWE) technique to assess the tissue stiffness of the cavernous bodies [42]. The measurements were conducted using the GE Logic S8 ultrasound machine (GE/General Electric, Chicago, IL, USA) equipped with a linear probe and using a 10 MHz frequency. The stiffness of the cavernous tissue was quantitatively expressed in KPa (kilopascal), and a color map was used to observe the qualitative stiffness of the tissue. The SWE study was performed before and after a 6-month period of treatment with a daily tablet containing a compound of Ecklonia bicyclis, Tribulus terrestris, and Biovis. 

The results of the study were as follows: Prior to treatment, the elastosonographic measurements showed that the average of three measurements (i.e., of the proximal, medial, and distal penis) was 37.05 KPa for the left cavernous body and 38.8 KPa for the right cavernous body. After six months of treatment with the compound, the examination revealed that the average fibrosity values were 31.07 KPa for the left cavernous body and 30.86 KPa for the right cavernous body, indicating a statistically significant decrease in fibrosity (*p* < 0.01). 

The authors concluded that SWE can be seen as an extension of the standard method used by doctors to assess the elastic tension and rigidity of the penis in an objective morphological examination.

#### 2.4.6. Tyloch et al. (2020) [43]

The authors conducted a study on 20 patients with PD using 2D and 3D ultrasound (US) [43]. All examinations involving US were conducted by a single experienced urologist using the B&K Medical Pro Focus 2202 (BK Medical, Herlev, Denmark) device with a 12 MHz linear probe. The first step in the study included normal, grayscale, axial, and sagittal US scans of the penis to visualize the PD plaque. The 3D ultrasound images were acquired in the second step. The length of time taken to acquire the 3D US images and perform the 2D US examination was measured for each case studied. In the 3D US assessment, the device captured separate US images as the probe moved, which were then converted into voxels (volume units) and reshaped into a 3D form (cube). After the examination, the cube was processed using computer software. The cavernous and spongy bodies were evaluated, and the plaque was identified and located, and its width, length, and thickness measured. A coronal view was obtained in addition to the standard axial and sagittal views, allowing for a comprehensive examination of the images. This included measuring the plaque, evaluating its location, and determining the number of plaques using the 3D cube obtained. The study yielded the following results: There was no significant difference in the average plaque size and surface area between the 3D US and 2D US measurements (127.72 mm^2^ vs. 128.74 mm^2^, *p* > 0.05). The use of the digital cube for image analysis reduced the average acquisition time to 69.8 s (median 64 s) for 3D US compared to 151.25 s (median 145.5 s) for 2D US (*p* < 0.05).

These results led the authors to state the following regarding 3D SWE: The 3D ultrasound exam only lasts as long as needed to gather the necessary data on the patient. The quantity and size of plaques were measured and evaluated after image acquisition, similar to CT and MRI, without the patient’s involvement. Imaging the entire plaque allows for a comparison of the images during treatment. 

Three-dimensional ultrasound examinations offer objective evaluations that can be reviewed by another physician during follow-up, which is challenging to achieve with two-dimensional ultrasound. The authors of the article concluded by stating that 3D US seems to be a valuable supplement to 2D US for PD patients. Furthermore, disease assessment using 3D US is completed after image capture and does not require patient participation. In addition to the fact that 3D US reduces the time needed for image acquisition, the process is made overall more comfortable for the patient.

#### 2.4.7. Trama et al. (2022) [44]

The authors carried out a study on 50 patients with PD (group A) and a control group of 50 healthy patients (group B) with the same characteristics regarding age, weight, and height [44]. The aim of this study was to compare the elasticity of the tunica albuginea (TA) of the penis in PD patients with that of control group subjects and to search for possible significant differences. Additionally, the study aimed to investigate the relationship between the stiffness of the penile corpora cavernosa and the severity of curvature, time to onset of curvature, and severity of penile pain during erection. Penile pain was assessed using the visual analog scale (VAS) with scores ranging from 0 (no pain) to 10 (highest perceived pain). The SWE was performed using the Logiq Healthcare Sq8 ultrasound system (GE/General Electric, Chicago, IL, USA); a single urology specialist with at least 3 years of experience in conducting SWE performed the examinations. 

The SWE procedure involved injecting 10 mcg of alprostadil into the corpora cavernosa to produce an erection. All patients completed the full International Index of Erectile Function (IIEF) questionnaire with 15 questions (IIEF-15), which evaluates various aspects of a patient’s sexuality, including erectile function, intercourse satisfaction, sexual desire, orgasmic function, and general satisfaction.

The study yielded the following findings: There were significant statistical differences in the mean values of the TA stiffness intensity (kPa) between the PD patient group (A) and the healthy group (B) (*p* < 0.0001). Additionally, the mean scores of the left and right TAs of groups A and B were calculated, revealing statistically significant differences (*p* < 0.0001). PD patients (group A) indicated that the average duration from diagnosis to curvature onset was 8.5 months (standard deviation: ±4.6).

The mean VAS score for patients with PD was 4.7 (standard deviation: ±2.6). Significant differences were found between the two groups in all domains of the IIEF-15 questionnaire (*p* < 0.0001), indicating that SWE was able to detect early tissue stiffness changes even in the absence of visible morphological features or detectable changes during a physical examination (e.g., non-palpable plaque).

The authors concluded the article with the following considerations: SWE is a non-expensive and noninvasive ultrasound method that could be useful in evaluating PD patients. Monitoring the progression of the condition following conservative treatments could be facilitated by measuring the stiffness of the tunica albuginea (TA) in kilopascals (kPa) (see Figure 3). Furthermore, SWE has the potential to enable the earlier detection of PD compared to the use of B-mode ultrasonography alone. There was a significant correlation between SWE and kPa values and curvature severity, duration from diagnosis to curvature onset, and pain level reported by patients, allowing for differentiation between the patients with PD and the control group.

#### 2.4.8. Zhao et al. (2024) [45]

The authors performed a study on 59 patients affected by PD to explore the diagnostic potential of shear-wave elastography (SWE) in diagnosing this condition [45]. 

The study also included a control group of 59 age-matched healthy adult men. 

B-mode US and SWE were performed on all PD patients by a single experienced radiologist using an AixplorerTM ultrasound device (Supersonic Imagine S.A., Aix-en-Provence, France) and a SuperLinearTM SL15-4 probe (frequency: 4–15 MHz). Both B-mode ultrasound (US) and SWE were conducted on all patients, and the Young’s modulus (YM) values of the respective penile areas in the PD and control groups were measured and subsequently compared. The study produced the following results: 41 out of the 59 included patients with PD (69.5%) were found to have penile plaques in the B-mode US evaluation, whereas the remaining 18 patients (30.5%) showed no evidence of penile plaques. After the SWE evaluation, it was discovered that the Young’s modulus (YM) values in penile plaque areas of the 41 PD patients had significantly higher stiffness values at 60.29 kPa compared to those outside the penile plaque area within the same subject at 21.05 kPa and in the corresponding penile area of the control group at 20.59 kPa (*p* < 0.001). Among the 18 remaining PD patients, the YM value of the abnormal penile area (56.67 kPa ± 13.52) was significantly higher than the YM value outside the abnormal area of the penis in the same patient (22.79 kPa) and in the same penile region in the control group (19.87 kPa) (*p* < 0.001; *p* < 0.001). The authors’ conclusion of the article was that SWE, as a noninvasive method, is effective at detecting and distinguishing penile plaques in patients with PD. It is a simple, fast, and complementary alternative to B-mode ultrasound.

#### 2.4.9. Richards et al. (2014) [46]

The authors published a case report of a patient suffering from PD with penile curvature who underwent shear-wave (SW) sonoelastography [46]. The purpose of the report was to describe a case of substantial penile curvature with a plaque that could not be visualized with sonography but was detectable with sonoelastography.

Case description: A 60-year-old man with atrial fibrillation and testosterone deficiency presented with a penile curvature to the left during erection. The curvature had persisted for 3 years, hindering his ability to have sexual intercourse. The patient did not mention any particular events that could have led to the penile curvature.

During the physical examination, his flaccid penis appeared normal, and no penile plaques were appreciated. B-mode and color Doppler ultrasound imaging of the penis with drug-induced erection revealed a 45° left curvature at the midshaft of the penis, without calcifications or ultrasound abnormalities observed in the tunica albuginea or corpora cavernosa. The images were detected with a 12 MHz linear array transducer SuperLinear^TM^ SL15-4 probe operating at a frequency range of 4–15 MHz with an ultrasound device capable of shear-wave sonoelastography (Aixplorer^®^, Supersonic Imagine S.A., Aix-en-Provence, France).

SW sonoelastography was used to examine the area, showing increased tissue stiffness at the level of the corpus cavernosum in the middle third of the penis, which matched the location of the maximum curvature. Transverse views displayed a firm area in the middle of the curvature on the left side of the penis, which was not visible in the images from the proximal or distal shaft. When the elastogram box was overlaid on the longitudinal penile images of the midshaft, firm tissue was only seen on the left side. This area involved both the tunica albuginea and the underlying cavernous tissue. Sonoelastography was crucial in confirming the diagnosis of PD, particularly when the physical examination and drug-induced duplex ultrasound did not show any evidence of a plaque.

The authors concluded the article with the following reflections: Penile sonoelastography is particularly beneficial for PD patients who do not have a detectable plaque through palpation or B-mode ultrasound. SW sonoelastography can be especially helpful in cases where intralesional treatment is required, providing an additional tool to locate a lesion for therapy when traditional ultrasonography cannot identify a plaque.

This option is especially important for this group of patients, considering the strong psychological impact of this disease and the high rates of patient dissatisfaction with more invasive techniques such as graft procedures [22,71,72]. Furthermore, SW sonoelastographic images have the potential to show tissue changes that go beyond what can be seen with a B-mode ultrasound. This means that sonoelastography can potentially demonstrate changes not only in the tunica albuginea but also in the cavernosal body.

#### 2.4.10. Dhawan et al. (2022) [47]

The authors published a case report of a patient suffering from PD with penile curvature who underwent ultrasound strain elastography [48]. The case described involved a 42-year-old male who had been experiencing penile curvature and ED for one year.

The patient’s ED manifested as difficulty in achieving or maintaining an erection. There was no prior history of trauma or urethral instrumentation.

Diagnostic evaluation: Linear calcific plaques with posterior acoustic shadowing were detected along the dorsal aspect of the penile shaft during the traditional ultrasound examination, affecting the anterior aspect of the corpora cavernosa.

Fibrotic tissue was observed on the dorsal side of the penis. Additionally, upon ultrasound strain elastography, a strain ratio of 3.8 was observed, visualized as a dark blue color suggestive of increased stiffness. Additional assessment with CT was conducted to pinpoint the exact location of the plaques and to detect any deeper plaques that may have been missed during the ultrasound examination.

The CT scan showed calcified plaques at the distal end of the penile shaft. In conclusion, the authors argued the following: Sonoelastography is a technique that can be used to assess non-palpable lesions that are not visible on ultrasonography. Furthermore, real-time strain elastography can be used to assess tissue elasticity. A variety of imaging techniques are employed to assess PD, but ultrasonography is the preferred modality due to its noninvasive nature and its ability to assess the severity of ED. It can also pinpoint the location and determine the thickness and size of a plaque, as well as detect any calcification within it. When combined with a patient’s clinical history and a physical examination that includes palpation of the penis, ultrasonography can help lead to a diagnosis of PD. 

For completeness, we would like to add that this case report did not report the name and type of the ultrasound machine used in this clinical case. 

#### 2.4.11. Parmar et al. (2020) [37]

This is a review of the general topic of “…imaging in the diagnosis and management of Peyronie’s disease” [37]. The article states that CT and radiography are excellent at visualizing penile plaque calcifications, and MRI is adept at identifying plaques in complex locations, such as in the corporal septum. Ultrasonography has a wide range of applications in localizing and characterizing plaques. However, more specifically, with regard to penile ultrasound imaging and elastography, the authors reported that high-resolution B-mode ultrasonography can be used as follows:-To assess the presence of concurrent ED to determine the need for a penile prosthesis for comprehensive treatment;-To determine whether plaques involve the neurovascular bundle or cavernosal artery, aiding in surgical preparation;-To conduct ultrasound assessments in the office, saving both time and money.

Moreover, penile Doppler ultrasound imaging not only serves as a diagnostic tool but also allows for monitoring treatment responses by tracking changes in plaque size, location, and calcification level over time.

In comparison to other modalities that have been discussed, penile Doppler ultrasound is the preferred first-line option due to its versatility, time efficiency, cost-effectiveness, lack of radiation risk, and ability to be used at the bedside. 

##### Sonoelastography

Ultrasound elastography, also known as sonoelastography, is a new addition to the ultrasonography repertoire that has shown potential in assessing PD and ED. Sonoelastography involves applying mechanical stress to tissue, which creates vibration patterns that can be used to identify variations in tissue elasticity. This method improves the ability to detect firm structures, like penile plaques, as these areas have lower elasticities than neighboring tissues.

There are various forms of sonoelastography, with the most frequently utilized being real-time SE and SWE. Real-time SE involves applying compression to the tissue with the ultrasound probe and analyzing the changes in vibration before and after compression to evaluate elasticity. SWE utilizes low-amplitude acoustic pulses from the probe to displace the tissue and create shear forces that allow elasticity to be measured. Sonoelastography is beneficial for evaluating non-palpable lesions related to PD that are not detectable through ultrasonography or other imaging techniques.

In their review, the authors also mention and comment on several articles that concern sonoelastography, explaining their contents [40,46]. 

The authors concluded their review as follows: There are several imaging methods that can be used to assess PD. Penile Doppler ultrasound is a noninvasive method used to evaluate the penis in PD patients. It can be used to assess ED severity; measure penile curvature; identify the location, size, and thickness of plaques; and detect calcification. Penile Doppler ultrasound is recommended in daily diagnostic and therapeutic approaches to PD. 

#### 2.4.12. Simon et al. (2022) [48]

These authors published a review article on the topic of elastography in urological practice, focusing on the urinary and male genital tract and excluding the prostate [48]. In this article, a search was conducted in the Cochrane Library and PubMed electronic databases. The authors found 94 articles of interest. We do not comment on the articles that were not of interest to our specific topic.

There were only two articles of interest regarding elastography as a diagnostic tool for Peyronie’s disease included in this review [40,46]. In a case report by Richards et al. (2014), 2D SWE was utilized to detect plaques and assist in intralesional injection treatment, as B-mode and Doppler ultrasonography did not reveal any evidence of plaques [46]. Riversi et al. used real-time elastosonography (RTE) to assess 74 patients with Peyronie’s disease symptoms. B-mode ultrasonography detected penile plaques in 64 patients, and RTE confirmed these results in addition to finding an additional five plaques in the remaining 10 patients [40].

Finally, the authors concluded that sonoelastography can be used in addition to two-dimensional and Doppler US to more accurately evaluate penile plaques even during the treatment of PD patients [53]. 

## 3. Discussion

The basic search in Medline for the term “ultrasound elastography” yielded 16,032 articles, whereas a search using the keywords “ultrasound elastography” and “Peyronie’s disease” resulted in 12 articles. It can be deduced that, unfortunately, the scientific literature on the use of elastography in Peyronie’s disease is relatively scarce and that ultrasound elastography is not widely known and practiced by uro-andrological specialists in the field.

The initial consideration of this review is that elastography is a relatively recent noninvasive technique that allows for the quick and easy evaluation of the actual mechanical properties of the tissue under examination, and it can help avoid invasive techniques such as biopsy (e.g., of the liver, prostate, etc.). 

We believe that strain elastography (SE), when combined with B-mode ultrasound and color Doppler imaging, can be easily performed and may offer valuable information. SE is not commonly performed in clinical practice for diagnosing and monitoring PD. In our review, we found four articles where this technique was used in the more in-depth diagnosis of PD. All of the authors that used this technique emphasized its clinical utility in identifying non-palpable Peyronie’s plaques and in evaluating plaque rigidity [38,39,40,47]. While the use of SE in combination with the analysis of semi-quantitative parameters may not be as quick as simple visual observation, it appears to offer more objective data and is better suited for tracking the progression of PD. It is important to obtain high-quality elastography images to ensure an accurate visual scan and to calculate reliable semi-quantitative data. It is recommended to use a dual-display (B-mode US display and elastography display) approach to ensure that the scanning plane is maintained during the real-time acquisition of elastography images. Since drawing the ROI is a procedure that depends on the operator, it is recommended that the operator is an expert in the field and has undergone a necessary period of diagnostic training involving patients with PD. However, for this elastographic method to have a significant impact on PD, it is necessary to standardize its use and conduct further studies to validate the results already documented in the literature.

Regarding transient elastography (TE), sometimes called 1D SWE, FibroScan uses dedicated equipment capable of measuring stiffness, but it does not combine reference ultrasound images to better localize the area to be examined [49]. It also has limited depth and encounters difficulties when examining obese patients. TE is primarily used for the investigation of fibrotic liver diseases, but it has also been applied in other medical contexts, such as in the examination of transplanted kidneys and breast cancer diagnosis [57,58,59,60,73,74]. However, new improvements have been made to FibroScan to make it more powerful and efficient [75]. In 2002, a group of researchers, including physicists and engineers, tried to create a new imaging method for use in tandem with TE/FibroScan. They experimented with a two-dimensional (2D) imaging device that, when used with TE, was able to produce an image [76,77]. Nevertheless, there have been no new practical developments and clinical applications since these experiments. Since TE only provides a one-dimensional (1D) quantitative representation (i.e., a line) of the tissue, the lack of image superimposition on an ultrasound image makes it impossible to use for the diagnostic evaluation of PD.

The ARFI technique is used to generate shear waves in various sonoelastography techniques. The pulse is designed specifically to cause mechanical movement in tissues of interest and create shear waves. Unlike TE, p-SWE and 2D SWE integrate imaging with elastography. Real-time imaging can be used to identify and avoid large masses and vascular structures.

Several studies have demonstrated that p-SWE is more reliable than TE and has comparable, if not superior, accuracy in detecting significant fibrosis [78,79,80,81]. In a meta-analysis, Bota et al. discovered that the failure rate in obtaining accurate evaluations was 2.1% for ARFI and 6.6% for TE [79].

Regarding supersonic shear imaging (SSI), Cassinotto et al. studied 349 patients suffering from chronic liver disease who had undergone liver biopsy, SSI, FibroScan, and ARFI to assess liver stiffness [82]. They found that SSI was more accurate than FibroScan in diagnosing severe fibrosis (*p* = 0.0016) and more accurate than ARFI in diagnosing significant fibrosis (*p* = 0.0003). There was no significant variance in diagnosing cirrhosis and mild fibrosis.

Regarding the clinical use of shear-wave elastography (SWE) in patients with PD, we found five articles in the literature where this method was used. The authors of these articles have indicated that this technique is very useful for identifying non-palpable or nondetectable plaques with simple ultrasound equipment, as well as for monitoring changes in plaque rigidity during and after conservative treatments [42,43,44,45,46]. 

In the literature, we found only one article that used vibro-elastography in the diagnosis of PD. However, this study was limited to investigating the specific changes in erectile tissue after the injection of vasoactive erectogenic drugs and did not study penile plaque and its diagnostic implications resulting from the use of vibro-elastography [41].

The search for literature “review” articles on the use of elastography as a diagnostic tool for PD resulted in only two articles [37,48]. Both reviews discussed the same studies that implemented the elastography method in PD patients, which we also found in our search and have already discussed in the present article. This leads us to the conclusion that, regrettably, despite being a valid imaging method for PD, ultrasound elastography is not being fully utilized to realize its potential as a diagnostic tool.

We believe that elastography is a very useful method for detecting Peyronie’s plaques, especially when they are not detectable with normal ultrasound or simply not palpable in the penis during a physical examination of the patient. Ultrasound elastography is also useful in studying the stiffness of the plaque and its variations during and after conservative treatments.

We would like to add that if the elastographic method allows for a more in-depth diagnosis of PD, there will probably be better therapeutic control of the disease. We know that this condition causes significant disruptions in the patient’s sexual life, both due to the morphological changes in the penis and the presence of pain, as well as a possible reduction in erectile potency. A more accurate diagnosis of PD will, therefore, make it possible to improve therapy with positive effects not only on penile deformation, pain, and erectile potency but also on the anxious–depressive state, which unfortunately represents a symptom with a notable impact on patients.

Out of the twelve articles we found in the Medline database using the keywords “ultrasound elastography” and “Peyronie’s disease”, six were published within the last 5 years. The first article on the topic was published 25 years ago, indicating a growing interest in elastography as a diagnostic method for Peyronie’s disease in the uro-andrological field. 

Although the topic was not the focus of our literature review, in this section, we believe it would be useful to expand the discussion by addressing the comparison between ultrasonography and other imaging techniques such as CT and MRI, which have emerged as other important tools in the diagnosis of Peyronie’s disease. We considered it appropriate to explore both the positive and negative aspects of the different imaging techniques used in the diagnosis of PD in order to discern which is the most suitable.

The only advantage of CT is its exceptional ability to visualize calcified plaques. In addition, the negative aspects of CT include its inadequate visualization of non-calcified plaques, soft tissues, thickening of the tunica albuginea, and the degree of inflammation. It also involves the use of ionizing radiation; is expensive in terms of money, time, and resources; and cannot be performed in the office. 

The strengths of MRI include its excellent soft tissue contrast that allows the visualization of the corpora cavernosa, fascial layers, and vascular system of the penis. It can also identify non-calcified plaques in complex locations such as the intercavernous septum and at the base of the penis, as well as active inflammatory zones, helping to determine appropriate treatment based on the stage of the disease. The negative aspects of MRI include that it is not very effective at detecting calcified plaques, it is quite expensive, it requires longer execution times and more resources, and it cannot be performed in the office.

The strengths of high-resolution penile echo-color Doppler imaging are that it is able to accurately detect plaques in most locations of the penile corpora cavernosa, including the septum; it can detect small plaques, sometimes even non-palpable ones; it is very useful for detecting calcified plaques without the use of ionizing radiation; it can be used to monitor response to treatments; and it can be performed in the office (at the bedside) with lower costs compared to CT and MRI and with faster execution times. However, the negative aspects of this technique are that its results are operator-dependent and that it sometimes poorly visualizes plaques in complex areas such as at the base of the penis.

The positive aspects of echo-elastography are as follows: it can identify non-palpable plaques and plaques that are not visible with traditional ultrasound and other imaging diagnostic techniques; it does not involve the use of ionizing radiation; and it can be performed in the office (at the bedside). However, the negative aspects of this method are that it is still in the experimental phase and would need additional research to assess its usefulness in clinical settings. Moreover, it is an operator-dependent test [37]. Although imaging techniques such as CT and MRI can be used in the diagnosis of PD, we believe that elastography associated with traditional ultrasound imaging is preferable to both of the above-mentioned techniques.

Andresen et al. conducted a study in which they examined and compared ultrasound sonography, mammography X-ray, CT, and MRI images from 20 PD patients. They concluded that high-resolution ultrasound sonography is the most effective imaging technique for evaluating plaques [83]. 

The main limitation of this study is that our review is simply a narrative review and not a systematic one (e.g., following PRISMA guidelines). Another limitation of our article is that both the article by Lahme et al. and the case report article by Dhawan et al. mentioned above do not explicitly mention the brands or models of the ultrasound equipment used in their studies [38,47]. This omission could be considered a limitation because knowing the exact device allows readers to better understand potential variations in image quality, performance, and reliability between different machines. Without this information, it becomes more difficult to accurately compare results between studies using different ultrasound systems.

Nevertheless, despite the limitations of our study, we believe that this article is very useful for urologists and andrologists to significantly improve their diagnostic capabilities in the context of Peyronie’s disease. It also highlights the importance of using elastography techniques to diagnose the disease when conventional methods have difficulty facilitating such a diagnosis.

## 4. Conclusions

Ultrasound elastography is an effective technique for identifying Peyronie’s plaques, particularly when they cannot be detected using standard ultrasound or are not palpable during a physical examination of the patient’s penis. Ultrasound elastography is useful for examining the stiffness of the plaque, but it is also useful to detect improvements, such as a quantitative reduction in the “deformation index” during and after conservative treatments. Elastography is not widely known or practiced in the field by uro-andrological specialists. Furthermore, solid experience in conventional ultrasound procedures is necessary to use elastography as effectively as possible. In addition, it is essential to have a strong background in traditional ultrasound techniques to maximize the effectiveness of elastography. Nevertheless, additional research is required to validate the efficacy of elastography in providing a comprehensive diagnosis of PD and to determine its superiority over traditional ultrasound techniques.

## 5. Future Directions

More research is needed in the form of larger, multicenter, prospective studies focused on the use of ultrasound elastography in PD. Furthermore, it is important to achieve a greater consensus on the complementary use of elastography techniques in the management of PD, not only for initial diagnosis but also during and after conservative treatments. Thorough training is necessary to improve the skills of uro-andrologist practitioners. Finally, they must be aware of the need for a more in-depth diagnosis of this disease, especially given the limited specific literature available to date, so that they can offer more targeted and effective treatment.

## Figures and Tables

**Figure 1 diagnostics-14-00665-f001:**
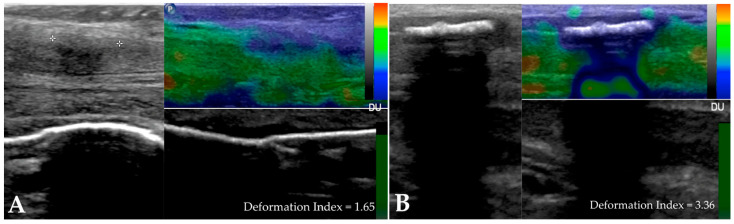
Strain elastography (SE) study performed on two patients with Peyronie’s disease: (**A**) penile plaque with a heterogeneous echostructure, isohypoechoic, without intralesional calcifications, and located in the distal third of the penis; (**B**) penile plaque located in the middle third of the penis, with a heterogeneous echostructure, isohyperechogenic, and with large intralesional calcifications measuring 13.6 × 5.6 mm.

**Figure 2 diagnostics-14-00665-f002:**
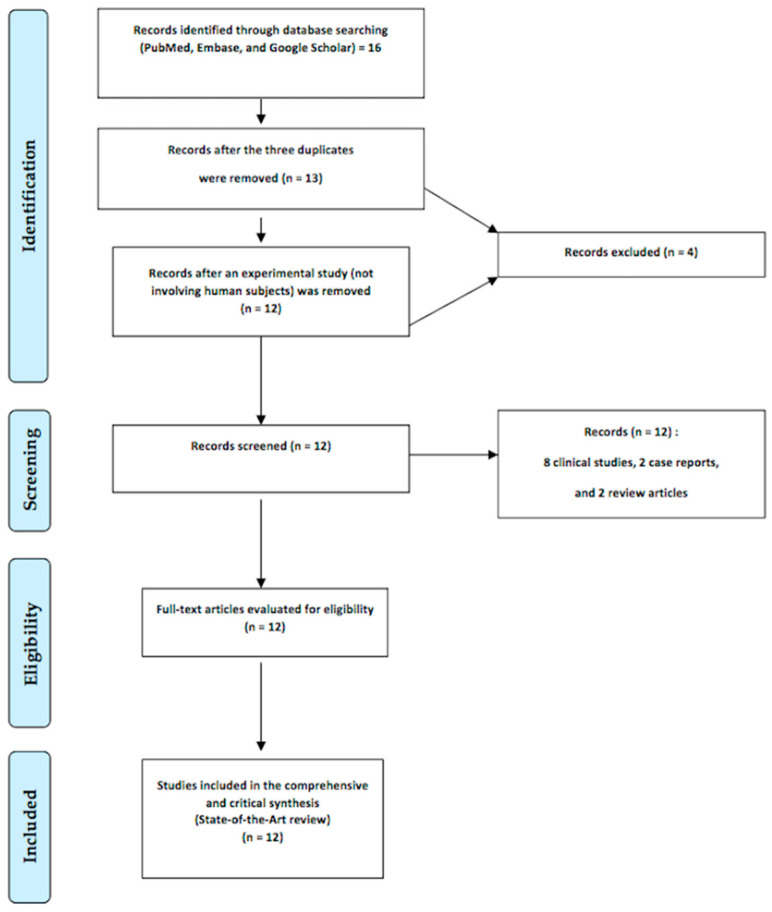
Flowchart of the state-of-the-art review of ultrasound elastography in the diagnosis of Peyronie’s disease.

**Figure 3 diagnostics-14-00665-f003:**
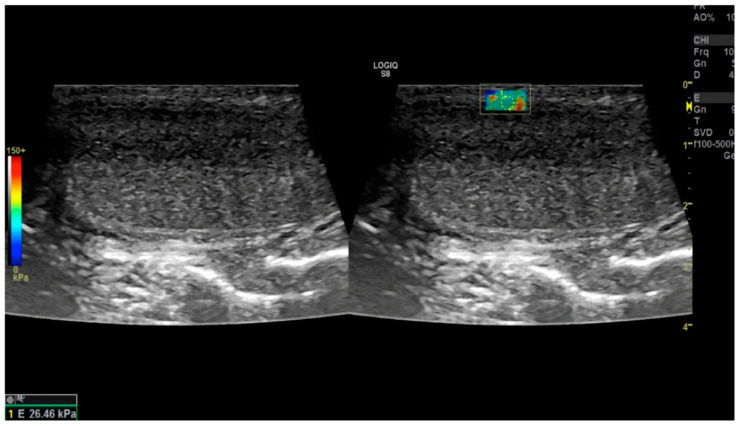
Use of penile shear-wave elastosonography for the diagnosis of Peyronie’s disease. Taken from the original article: Use of penile shear wave elastosonography for the diagnosis of Peyronie’s Disease: A prospective case-control study [44]. «Fig. 2 Example of a shear wave elastography image of Tunica Albuginea (TA) including quantitative measurement data expressed in kilopascal (kPa) in a patient with Peyronie’s Disease (PD) (case group). The elastosonography mode simultaneously provides two images of the same area: The left image is in US—B—mode while on the right image is simultaneously displayed for stiffness measurement. The yellow box indicates the region of interest (ROI) observed in the shear wave elastography image, and the yellow circles indicate the intensity of the stiffness expressed values in kPa. In addition, the software provides a colorimetry map of the stiffness of the ROI. In particular, the red color indicates high stiffness while the blue color indicates lower stiffness. TA: tunica albuginea; kPa: kilopascal; PD: Peyronie’s Disease US: ultrasound; ROI: region of interest». © Copyright Policy—Open access, License Creative Commons. Source of image: [44]. Link: https://bacandrology.biomedcentral.com/articles/10.1186/s12610-022-00164-w (accessed on 16 March 2024).

## Data Availability

Not applicable.

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
