# Peer review of "Ultrasound Elastography as a Diagnostic Tool for Peyronie’s Disease: A State-of-the-Art Review"

_diagnostics, 2024, doi:10.3390/diagnostics14060665_

Round 1

Reviewer 1 Report

Comments and Suggestions for Authors

The current study aims to provide a better understanding of the diagnosis of PD using US elastography.

The study investigated in the literature, and provides an overview of the literature to guide further research in this field and stimulates uro/andrologists to adopt this methodology.

The authors should be congratulated for the work and for addressing an interesting topic. Only a few points warrant mentions:

1.      Please define PD also in the text (line 34)

2.      In line 34, the declaration that PD has a genetic origin is hastened, there is only a genetic predisposition, at later stated in line 38. Please correct.

3.      In the “Introduction” section, I suggest that the authors also show data on the different burdens of the disease according to patients’ age, such as in PMID: 36426559

4.      In the “Introduction” section, I suggest ending the section reporting the aim of the study by moving it from “2.4 Literature review” section. Moreover, it is not completely clear. Why did the authors perform this review? Can elastography give some advantages in the diagnosis of the physical properties of the PD plaque? Can its properties change the management of the disease? Please, address this point.

5.      Avoid defining an acronym again after doing it before (i.e., line 123, region of interest (ROI); erectile dysfunction and Peyronie’s disease in lines 443-444).

6.      Line 355, I suggest to modify “Our results”.

7.      How did the authors report data from selected studies? I suggest to follow at least a chronological order of publishing.

8.      An English language revision is required. Some sentences are misunderstandable.

Comments on the Quality of English Language

English style should be revised.

Author Response

Dear Colleague,

Thank you very much for taking the time to review this manuscript.

  1. Please define PD also in the text (line 34)
  2. In line 34, the declaration that PD has a genetic origin is hastened, there is only a genetic predisposition, at later stated in line 38. Please correct.

Answer

  1. Dear Colleague, thank you for your suggestion. I added new text to the sentence.
  2. Dear colleague, thank you for noticing this. I corrected the incorrect sentence as you rightly suggested. The deleted text was highlighted in red. The corrected text was written in green.

---------------------

  1. In the “Introduction” section, I suggest that the authors also show data on the different burdens of the disease according to patients’ age, such as in PMID: 36426559

Answer

Dear Colleague, thank you for your suggestion. I added new text to the sentence.     

I have added the new citations to the bibliography. The added text was written in green.

Necessarily the numbering of the citationss was changed and the new numbering in the text was written in green.

 ---------------------

  1. In the “Introduction” section, I suggest ending the section reporting the aim of the study by moving it from “2.4 Literature review” section. Moreover, it is not completely clear. Why did the authors perform this review? Can elastography give some advantages in the diagnosis of the physical properties of the PD plaque? Can its properties change the management of the disease? Please, address this point.

Answer

Dear Colleague, thank you for your suggestion. I moved the sentence you pointed out to the end of the Introduction section. Additionally, I also added the main reasons for why we conducted this literature review. The deleted text was highlighted in red. The moved text and the added one was written in green.

---------------------

  1. Avoid defining an acronym again after doing it before (i.e., line 123, region of interest (ROI); erectile dysfunction and Peyronie’s disease in lines 443-444).

Answer

Dear Colleague, thank you for your suggestion. I corrected the text as you suggested, always writing the acronym after having already described it previously. The deleted text was highlighted in red. The corrected text was written in green.

---------------------

  1. Line 355, I suggest to modify “Our results”.

Answer

Dear Colleague, thank you for notifying me of the error. I corrected the text. The deleted text was highlighted in red. The corrected text was written in green.

---------------------

  1. How did the authors report data from selected studies? I suggest to follow at least a chronological order of publishing.

Answer

Dear colleague, thank you for your suggestion. However, I would like to clarify, as I have already done in the manuscript on lines 338-340, that the selected studies were sorted in ascending order by year of publication and type, in the following order: 1) studies on cohorts of patients with PD, 2) case reports, and 3) reviews. In this way, the selected studies have already been sorted by year of publication, while still respecting the typology of the selected study.

 ---------------------

  1. An English language revision is required. Some sentences are misunderstandable.

Answer

Dear Colleague, I thank you for your suggestion, but I want to inform you that before submitting the article to the Diagnostics Journal, I used the MDPI-English editing Service to have the manuscript checked and corrected in correct English from the native English language.

However, I had my current revision proofread again by the MDPI-English editing certificate. I attach the MDPI-English editing certificate.

Dear Colleague, thank you again for dedicating your time to review my manuscript.                                                                                                                    Note that the manuscript was also changed in relation to the requests of the second reviewer.                                                                                                   Kind regards

Reviewer 2 Report

Comments and Suggestions for Authors

Dear authors,

The articleUltrasound Elastography as a Diagnostic Tool for Peyronie’s Disease: A State-of-the-Art Review' offers an insightful perspective into the use of sonoelastography and other diagnostic tools for evaluating Peyronie's disease (PD). While the paper presents valuable information about the application of these methods, I will focus on one specific critique related to methodological detail and completeness.

The article does not explicitly mention the brand or model of the ultrasound equipment utilized in the clinical case described. This omission could be considered a limitation because knowing the exact device allows readers to better understand potential variations in image quality, performance, and reliability across different machines. Without this information, it becomes more difficult to accurately compare findings between studies using various ultrasound systems. Please add this idea in the limitation section.

Additionally, while the study highlights the advantages of ultrasonography over other modalities in diagnosing PD, it does not provide data comparing ultrasonography directly against alternative imaging techniques such as magnetic resonance imaging (MRI), which has emerged as another important tool in PD diagnostics. Such comparative analysis might further strengthen the argument for ultrasonography being the preferred modality. Please include a paragraph in the Discussion section.

Based on the iThenticate report, which indicates a percent match of 31%, it's important to carefully review the areas where similarities are found. While a 31% match may not necessarily indicate plagiarism or misconduct, it does warrant attention to ensure that proper citation practices are followed and that any quoted or paraphrased material is appropriately attributed. Here are some steps to consider:

  1. Review the Report: Take the time to thoroughly review the iThenticate report to identify the specific areas of similarity and understand the context in which they occur.
  2. Evaluate Similarities: Determine whether the similarities are due to common phrases, quotations, citations, or potentially problematic instances of unattributed content.
  3. Address Citations: Ensure that all quoted material is properly cited according to the appropriate citation style (MDPI guidelines). Verify that paraphrased content is also appropriately attributed and does not rely too heavily on the original wording.
  4. Rephrase or Quote: If necessary, consider rephrasing passages to reduce similarity, or if quoting directly is necessary, ensure proper quotation marks and citation of the original source.
  5. Follow Institutional Policies: Adhere to the plagiarism policies and guidelines established by your institution or organization when addressing any instances of similarity.

By taking these steps, you can effectively manage and address the percent of similarities identified in the iThenticate report, promoting academic integrity and responsible research practices.

Overall, however, the article provides a comprehensive overview of sonoelastography and its role in the diagnosis of PD, emphasizing its utility in assessing nonpalpable lesions and offering insights into real-time strain elastography for measuring tissue elasticity. After revision of the content, the inclusion of additional imaging techniques and their relative strengths and weaknesses in diagnosing PD adds value to the discussion.

Comments on the Quality of English Language

Kindly correct some minor English mistakes.

Author Response

Comments and Suggestions for Authors

Dear authors,

The articleUltrasound Elastography as a Diagnostic Tool for Peyronie’s Disease: A State-of-the-Art Review' offers an insightful perspective into the use of sonoelastography and other diagnostic tools for evaluating Peyronie's disease (PD). While the paper presents valuable information about the application of these methods, I will focus on one specific critique related to methodological detail and completeness. Overall, however, the article provides a comprehensive overview of sonoelastography and its role in the diagnosis of PD, emphasizing its utility in assessing nonpalpable lesions and offering insights into real-time strain elastography for measuring tissue elasticity.

Answer

Dear Colleague, Thank you very much for taking the time to review this manuscript.

Thank you for all your suggestions.

---------------------

The article does not explicitly mention the brand or model of the ultrasound equipment utilized in the clinical case described. This omission could be considered a limitation because knowing the exact device allows readers to better understand potential variations in image quality, performance, and reliability across different machines. Without this information, it becomes more difficult to accurately compare findings between studies using various ultrasound systems. Please add this idea in the limitation section.

Answer

Dear colleague, thank you for your suggestion.

As you suggested, I have added some considerations in the limitations-section of the Discussion.

I cited the two articles (including Lhame's study) where the brand or model of ultrasound machine used was not mentioned.

In my article I also added the mention of the ultrasound machine that we used in the two cases described in Figure 1.The added texts (sentences) were written in green.

---------------------

Additionally, while the study highlights the advantages of ultrasonography over other modalities in diagnosing PD, it does not provide data comparing ultrasonography directly against alternative imaging techniques such as magnetic resonance imaging (MRI), which has emerged as another important tool in PD diagnostics. Such comparative analysis might further strengthen the argument for ultrasonography being the preferred modality. Please include a paragraph in the Discussion section.

Answer

Dear colleague, thank you for your suggestion.

As you suggested, I have added in the discussion section a comparison between ultrasound and other alternative imaging techniques, and also the strengths and weaknesses of the various imaging techniques used in the diagnosis of PD. The added text (sentence) were written in green.

---------------------

Based on the iThenticate report, which indicates a percent match of 31%, it's important to carefully review the areas where similarities are found. While a 31% match may not necessarily indicate plagiarism or misconduct, it does warrant attention to ensure that proper citation practices are followed and that any quoted or paraphrased material is appropriately attributed. Here are some steps to consider:

  1. Review the Report: Take the time to thoroughly review the iThenticate report to identify the specific areas of similarity and understand the context in which they occur.
  2. Evaluate Similarities: Determine whether the similarities are due to common phrases, quotations, citations, or potentially problematic instances of unattributed content.
  3. Address Citations: Ensure that all quoted material is properly cited according to the appropriate citation style (MDPI guidelines). Verify that paraphrased content is also appropriately attributed and does not rely too heavily on the original wording.
  4. Rephrase or Quote: If necessary, consider rephrasing passages to reduce similarity, or if quoting directly is necessary, ensure proper quotation marks and citation of the original source.
  5. Follow Institutional Policies: Adhere to the plagiarism policies and guidelines established by your institution or organization when addressing any instances of similarity.

By taking these steps, you can effectively manage and address the percent of similarities identified in the iThenticate report, promoting academic integrity and responsible research practices.

Answer

Dear colleague, thank you for your suggestion.

As you suggested, I have corrected the manuscript, significantly reducing the percentage of plagiarism detected. Due to the short deadline for my review (unfortunately I did not receive a further delay in the deadline date), I was not able to revise the entire manuscript for plagiarism, but I believe that my corrections are sufficient to make the manuscript with non-significant plagiarism percentages for publication.

---------------------

Dear Colleague, thank you again for dedicating your time to review my manuscript.                                                                                                                    Note that the manuscript was also changed in relation to the requests of the first reviewer.                                                                                                             Kind regards

Round 2

Reviewer 1 Report

Comments and Suggestions for Authors

I thank the authors so much for addressing all my comments and explaining some concernings.

Reviewer 2 Report

Comments and Suggestions for Authors

Dear authors,

The revised version of the article 'Ultrasound Elastography as a Diagnostic Tool for Peyronie’s Disease: A State-of-the-Art Review' presents the changes required.

The article offers an insightful perspective on the use of sonoelastography and other diagnostic tools for evaluating Peyronie's disease (PD). 

I recommend thoroughly reviewing the entire manuscript and rectifying any English writing issues.

Comments on the Quality of English Language

I recommend thoroughly reviewing the entire manuscript and rectifying any English writing issues.